# ADAPTIVE GUIDANCE SCALING FOR POSTERIOR DIFFUSION-BASED SAMPLING

## ABSTRACT

Diffusion models have recently emerged as powerful generative priors for solving inverse problems, achieving state-of-the-art results across various imaging tasks. A central challenge in this setting lies in balancing the contribution of the prior with the data fidelity term: overly aggressive likelihood updates may introduce artifacts, while conservative updates can slow convergence or yield suboptimal reconstructions. In this work, we propose an adaptive likelihood step-size strategy to guide the diffusion process for inverse-problem formulations. Specifically, we develop an observation-dependent weighting scheme based on the agreement between two different approximations of the intractable intermediate likelihood gradients, that adapts naturally to the diffusion schedule, time re-spacing, and injected stochasticity. The resulting approach, **A**daptive **P**osterior diffusion **S**ampling (APS), is hyperparameter-free and improves reconstruction quality across diverse imaging tasks—including super-resolution, Gaussian deblurring, and motion deblurring—on CelebA-HQ and ImageNet-256 validation sets. APS consistently surpasses existing diffusion-based baselines in perceptual quality without any task-specific tuning. Extensive ablation studies further demonstrate its robustness to the number of diffusion steps, observation noise levels, and varying stochasticity.

## 1 INTRODUCTION

Image restoration arises in numerous applications, where the goal is to recover a high-quality image $x \in \mathbb{R}^n$ from a degraded observation $y \in \mathbb{R}^m$ that may be noisy, blurry, low-resolution, or otherwise corrupted. In many cases, the relationship between $y$ and $x$ can be modeled as

$$y = \mathcal{A}(x) + \varepsilon, \tag{1}$$

where $\mathcal{A} : \mathbb{R}^n \to \mathbb{R}^m$ is a measurement operator, and $\varepsilon$ denotes additive noise (typically modeled as white Gaussian noise $\mathcal{N}(0, \sigma_y^2 I)$). For instance, in image denoising $\mathcal{A}$ is the identity operator; in deblurring, $\mathcal{A}$ represents a blur kernel; and in super-resolution, $\mathcal{A}$ consists of a composition of sub-sampling and anti-aliasing filtering.

Inverse problems of the form Eq. 1 are typically *ill-posed*: the solution may be nonunique (e.g., when $\mathcal{A}$ is not injective), unstable to perturbations in $y$ (e.g., when $\mathcal{A}$ is ill-conditioned), or may not exist without additional regularity assumptions. These challenges are particularly pronounced in underdetermined settings with $m \ll n$, where no exact inverse exists, and in the presence of measurement noise. Consequently, simply fitting the observation model does not guarantee accurate recovery, and incorporating prior knowledge about the structure of $x$ is essential.

A widely adopted paradigm is to train deep neural networks (DNNs) for each specific observation model. That is, synthetic training pairs $\{(y_i, x_i)\}$ are generated using Eq. 1, and a DNN is trained to approximate the inverse map (Dong et al., 2015; Lim et al., 2017; Sun et al., 2015; Zhang et al., 2017a). However, these task-specific networks typically suffer severe performance degradation when the test-time observations deviate, even slightly, from the training assumptions (Hussein et al., 2020; Shocher et al., 2018; Tirer & Giryes, 2019), limiting their practicality.

An alternative line of work leverages pretrained DNNs that capture only the signal prior, while consistency with the observations is enforced during inference in a "zero-shot" manner. A particularly

successful choice has been Gaussian denoisers, employed in "plug-and-play" (PnP) and "regularization by denoising" (RED) frameworks (Romano et al., 2017; Tirer & Giryes, 2018; Venkatakrishnan et al., 2013; Zhang et al., 2017b). The recent emergence of diffusion/score-based generative models (Ho et al., 2020; Song & Ermon, 2019; Song et al., 2020b) has further popularized iterative denoising for general-purpose restoration. In diffusion models, inference involves reversing a diffusion process by iteratively removing Gaussian noise until a clean sample is obtained. Explicit data fidelity terms have been integrated into this iterative sampling to ensure reconstructions that both appear natural and conform to the measurements (Abu-Hussein et al., 2022; Chung et al., 2022; 2023; Kawar et al., 2022; Mardani et al., 2023; Song et al., 2023; 2021; Wang et al., 2023; Zhu et al., 2023; Garber & Tirer, 2024).

Although utilizing the strong data prior offered by diffusion models has shown great results, forcing the data fidelity for guiding the diffusion trajectory to a reconstruction that agrees with the observation is usually performed by incorporating an approximation of the likelihood gradient into the sampling scheme, while compensating for the inaccuracy introduced to the sampling by carefully fine-tuning certain hyperparameters or imposing an additional projection onto the diffusion manifold (Kawar et al., 2022; Chung et al., 2023; Zhu et al., 2023; Garber & Tirer, 2024; Yang et al., 2024). Furthermore, this data fidelity step usually has fixed or pre-defined step sizes that depend on the diffusion noise scheduling parameters.

In this work, we introduce **A**daptive **P**osterior diffusion **S**ampling (APS), which addresses the core challenge of balancing prior and data fidelity contributions. Overly aggressive likelihood updates may introduce artifacts, while conservative updates can lead to slow convergence or suboptimal reconstructions. To overcome this, APS employs a novel weighting strategy that adaptively tunes the step size based on the agreement between two complementary approximations of the intractable intermediate likelihood gradients. This adaptive mechanism allows the sampling trajectory to flexibly adjust to the ill-posedness of the observation model, measurement noise level, and characteristics of the diffusion process, improving both robustness and reconstruction quality.

We evaluate APS on a variety of inverse problems, including super-resolution, Gaussian deblurring, and motion deblurring, across the popular CelebA-HQ and ImageNet-256 datasets. Our experiments demonstrate that APS consistently outperforms existing diffusion-based approaches in terms of reconstruction quality and scalability. Extensive ablation studies further confirm the effectiveness of the adaptive guidance mechanism with respect to the number of diffusion steps, observation noise level, and the stochasticity of the diffusion process.

Our main contributions are summarized as follows:

(1) **DDIM reformulation for conditional guidance.** We provide a principled way to incorporate likelihood gradients into an existing DDIM sampler via a conditional noise estimator. The resulting update preserves DDIM's scheduling and therefore naturally scales with the number of steps, time re-spacing, and the level of stochasticity.

(2) **Adaptive, hyperparameter-free guidance scaling.** Viewing posterior sampling through the lens of step-size selection, we introduce a data-dependent, hyperparameter-free rule that modulates the guidance by the agreement between two complementary surrogates. This yields robust, alignment-aware updates at negligible runtime cost.

(3) **Perceptually leading performance with competitive distortion.** APS achieves best or second-best LPIPS across tasks while maintaining strong PSNR—contrasting with baselines that optimize one metric at the expense of the other.

## 2 BACKGROUND

### 2.1 INVERSE PROBLEMS

In this work we focus on the widely studied *linear–Gaussian* case. The forward operator is linear, $\mathcal{A}(x) = Ax$ with $A \in \mathbb{R}^{m \times n}$, and the noise is modeled as zero-mean i.i.d. Gaussian with known variance $\sigma_y^2$, i.e., $\varepsilon \sim \mathcal{N}(0, \sigma_y^2 I_m)$, where $I_m \in \mathbb{R}^{m \times m}$ is the identity matrix.

From a Bayesian perspective, we treat $x \sim p(x)$ as a random unknown vector to be estimated from the observation $y$. A natural first step is to maximize the likelihood probability density function,

which has the form

$$p(y|x) \propto \exp\left( - \frac{1}{2\sigma_y^2} \|Ax - y\|_2^2 \right). \tag{2}$$

However, relying solely on the likelihood often yields unsatisfactory reconstructions due to the inherent instability and non-uniqueness of the solution. A more effective strategy is to maximize the posterior distribution,

$$p(x|y) \propto p(y|x)p(x), \tag{3}$$

which combines the likelihood with a prior $p(x)$. The inclusion of the prior greatly improves reconstruction quality, but also raises a central challenge: designing a suitable prior. The more accurately $p(x)$ captures the structure of the true signal, the more reliable the reconstruction will be.

## 2.2 Diffusion Models

Diffusion models are a class of generative models that synthesize data by *reversing* a gradual noising process (Ho et al., 2020; Song et al., 2020b). Both the forward (noising) and reverse (denoising) dynamics can be formalized with stochastic differential equations (SDEs). The forward process progressively corrupts clean data so that, at a terminal time $t = T$, the distribution becomes tractable—typically close to Gaussian. Thus, generating a novel data point amounts to solving the corresponding reverse-time dynamics to transport noise back to the data distribution.

**Forward and reverse dynamics.** For $t \in [0, T]$, let $x(t) \in \mathbb{R}^n$ evolve under the forward SDE

$$\mathrm{d}x = f(x, t)\, \mathrm{d}t + g(t)\, \mathrm{d}w, \tag{4}$$

where $f$ is the drift, $g(t) \geq 0$ is a scalar diffusion schedule, and $w$ is standard Brownian motion. The schedule is chosen such that, approximately, $x(T) \sim \mathcal{N}(0, I)$. The corresponding reverse-time SDE (Anderson, 1982), which shares the same time-marginals, is given by

$$\mathrm{d}x = \left[ f(x, t) - g(t)^2\, \nabla_x \log p_t(x) \right] \mathrm{d}t + g(t)\, \mathrm{d}\bar{w}, \tag{5}$$

where $p_t$ is the probability density function of $x(t)$ and $\bar{w}$ is a reverse-time Brownian motion. Solving Eq. 5 with respect to $x(t)$ requires access to the score function $\nabla_x \log p_t(x)$, which is unknown and must be approximated.

**VP–DDPM.** A widely used choice is $f(x, t) := -\frac{1}{2}\beta(t)\, x$ and $g(t) := \sqrt{\beta(t)}$, known as the variance-preserving (VP) parameterization in DDPM (Ho et al., 2020). In practical discrete settings, we denote $x(t) := x_t$ and $\beta(t) := \beta_t$. The forward diffusion kernel is then

$$p(x_t \mid x_0) = \mathcal{N}\left( x_t;\, \sqrt{\bar{\alpha}_t}\, x_0,\, (1 - \bar{\alpha}_t)I \right), \qquad \alpha_t := 1 - \beta_t, \quad \bar{\alpha}_t := \prod_{s=1}^{t} \alpha_s.$$

Equivalently,

$$x_t = \sqrt{\bar{\alpha}_t}\, x_0 + \sqrt{1 - \bar{\alpha}_t}\, \epsilon, \qquad \epsilon \sim \mathcal{N}(0, I). \tag{6}$$

We follow this framework and notation throughout the paper.

**Evaluating the score.** Diffusion models are trained to denoise the degraded signal $x_t$ by predicting either the clean signal $\hat{x}_0$ or the noise. In the latter case, let $\epsilon_\theta(x_t, t)$ denote the predicted noise at time $t$ using a DNN with parameters $\theta$. Prior works show that this predictor yields a score estimate. Specifically, for the VP–DDPM parameterization (Eq. 6) we have (Efron, 2011; Hyvärinen & Dayan, 2005; Vincent, 2011; Song et al., 2020b):

$$\nabla_{x_t} \log p_t(x_t) \approx -\frac{1}{\sqrt{1 - \bar{\alpha}_t}}\, \epsilon_\theta(x_t, t), \tag{7}$$

which follows from $\nabla_{x_t} \log p_t(x_t) = \frac{1}{1 - \bar{\alpha}_t}(\mathbb{E}[\sqrt{\bar{\alpha}_t}\, x_0 | x_t] - x_t) = -\frac{1}{\sqrt{1 - \bar{\alpha}_t}} \mathbb{E}[\epsilon | x_t]$, given by Tweedie's formula (Efron, 2011). Based on the relation in Eq. 6, the denoised signal can be obtained by

$$\hat{x}_0(x_t, t) = \frac{x_t - \sqrt{1 - \bar{\alpha}_t}\, \epsilon_\theta(x_t, t)}{\sqrt{\bar{\alpha}_t}}. \tag{8}$$

**Sampling.** Given $\epsilon_\theta(x_t, t)$, samples can be generated by numerically solving Eq. 5 using the approximation in Eq. 7. A common sampling algorithm is DDIM (Song et al., 2020a), where each intermediate sample $x_{t-1}$ is obtained by

$$x_{t-1} = \sqrt{\bar{\alpha}_{t-1}}\, \hat{x}_0(x_t, t) + \sqrt{1 - \bar{\alpha}_{t-1} - \sigma_t^2}\epsilon_\theta(x_t, t) + \sigma_t\, \epsilon_t, \qquad \epsilon_t \sim \mathcal{N}(0, I). \tag{9}$$

The stochasticity of the update is governed by $\sigma_t$, which is commonly parameterized by $\eta \in [0, 1]$, as $\sigma_t = \eta\, \sqrt{1 - \alpha_t}\, \sqrt{\frac{1 - \bar{\alpha}_{t-1}}{1 - \bar{\alpha}_t}}$. Thus, $\eta$ represents the level of stochasticity of the diffusion process.

## 2.3 POSTERIOR SAMPLING

Posterior sampling methods aim to draw samples from the conditional distribution $p(x_0|y)$ by constructing the *conditional score* $\nabla_{x_t} \log p_t(x_t|y)$ and integrating the reverse dynamics using it. From Bayes' rule,

$$\nabla_{x_t} \log p_t(x_t|y) = \nabla_{x_t} \log p_t(x_t) + \nabla_{x_t} \log p_t(y|x_t), \tag{10}$$

where the left-hand side is the *posterior score*, the first term on the right is the *prior score*, and the second is the *likelihood score*.

While the prior term can be obtained using an unconditional score network similar to Eq. 7, the likelihood term $\nabla_{x_t} \log p_t(y|x_t)$ is generally intractable. Specifically, using the law of total probability with $y \perp x_t$ given $x_0$, we have

$$p(y|x_t) = \int p(y|x_0)\, p(x_0|x_t)\, \mathrm{d}x_0. \tag{11}$$

The measurement model $p(y|x_0)$ is available from Eq. 2, but $p(x_0|x_t)$ is unknown. We next describe two common approximations to $p(x_0|x_t)$ that yield practical likelihood-score surrogates.

**DPS.** Chung et al. (2023) suggests to approximate $p(x_0|x_t) \approx \delta(x_0 - \hat{x}_0)$, where $\delta(\cdot)$ is the Dirac delta distribution. By Eq. 11, the likelihood score is then approximated by $\nabla_{x_t} \log p_t(y|x_t) \approx \nabla_{x_t} \log p\big(y|\hat{x}_0(x_t, t)\big) = -\sigma_y^{-2}(\frac{\partial \hat{x}_0}{\partial x_t})^\top A^\top (y - A\hat{x}_0)$ which can be obtained via backpropagation.

**ΠGDM.** Alternatively, Song et al. (2023) suggest to approximate $p(x_0|x_t)$ as Gaussian of the form $p(x_0|x_t) \approx \mathcal{N}(\hat{x}_0, r_t^2 I)$, with $r_t^2 = 1 - \bar{\alpha}_t$ (in VP-DDPM parameterization), which yields the surrogate likelihood score $\nabla_{x_t} \log p_t(y|x_t) \approx (\frac{\partial \hat{x}_0}{\partial x_t})^\top A^\top (r_t^2 AA^\top + \sigma_y^2 I)^{-1}(y - A\hat{x}_0)$.

Most posterior sampling methods incorporate these score terms into the DDIM update in Eq. 9 and weight them either heuristically (Song et al., 2023) or by tuning additional hyperparameters (Chung et al., 2023; 2022). In this work, we address the challenge of integrating the likelihood score into DDIM update in a balanced, scalable and robust way.

## 3 METHOD

### 3.1 REFORMULATING DDIM FOR CONDITIONAL SETTINGS

Substituting Eq. 8 into the DDIM update Eq. 9, we can write it in a Markovian form:

$$x_{t-1} = \frac{1}{\sqrt{\alpha_t}}\, x_t - \underbrace{\left( \frac{\sqrt{1 - \bar{\alpha}_t}}{\sqrt{\alpha_t}} - \sqrt{1 - \bar{\alpha}_{t-1} - \sigma_t^2} \right)}_{\gamma_t} \epsilon_\theta(x_t, t) + \sigma_t\, \epsilon_t =: \mathrm{DDIM}(x_t), \tag{12}$$

where $\epsilon_t \sim \mathcal{N}(0, I)$ and $\gamma_t$ collects time-dependent coefficients.

Building on the identity $\mathbb{E}[\epsilon|x_t, y] = -\sqrt{1 - \bar{\alpha}_t}\, \nabla_{x_t} \log p_t(x_t|y)$, which is a straightforward generalization of Tweedie's formula to the conditional case (see, e.g., Lemma A.2 in Peng et al. (2024)), we introduce a *posterior* surrogate $\tilde{\epsilon}_\theta(x_t, t, y)$ related in the same way to the *posterior* score $\nabla_{x_t} \log p_t(x_t|y)$. Using Eq. 10,

$$\tilde{\epsilon}_\theta(x_t, t, y) := \epsilon_\theta(x_t, t) + \xi_t\, g(y, x_t), \tag{13}$$

where $g(y, x_t)$ is any tractable estimator of the likelihood-score term $\nabla_{x_t} \log p_t(y|x_t)$ and $\xi_t \in \mathbb{R}$ balances the (approximate) likelihood and prior scores and encapsulates all derived constants. Plugging Eq. 13 into Eq. 12 (in lieu of $\epsilon_\theta$) yields the conditional DDIM step

$$x_{t-1} = \frac{1}{\sqrt{\alpha_t}} x_t - \gamma_t \epsilon_\theta(x_t, t) - \gamma_t \xi_t \ g(y, x_t) + \sigma_t \epsilon_t$$
$$= \text{DDIM}(x_t) - \gamma_t \xi_t \ g(y, x_t). \tag{14}$$

This formulation makes explicit that posterior information affects both the intermediate estimate $\hat{x}_0(x_t, t)$ (implicitly, through $\epsilon_\theta$) and the projection back to $t-1$ via an effective noise that interpolates the predicted noise and fresh randomness. In contrast, most methods directly add $\xi_t \, g(y, x_t)$ to DDIM, without explicitly accounting for the factor $\gamma_t$ (Chung et al., 2023; Song et al., 2023), while others heuristically modify DDIM's injected noise estimate (Garber & Tirer, 2024; Zhu et al., 2023).

## 3.2 Adaptive Posterior Sampling (APS)

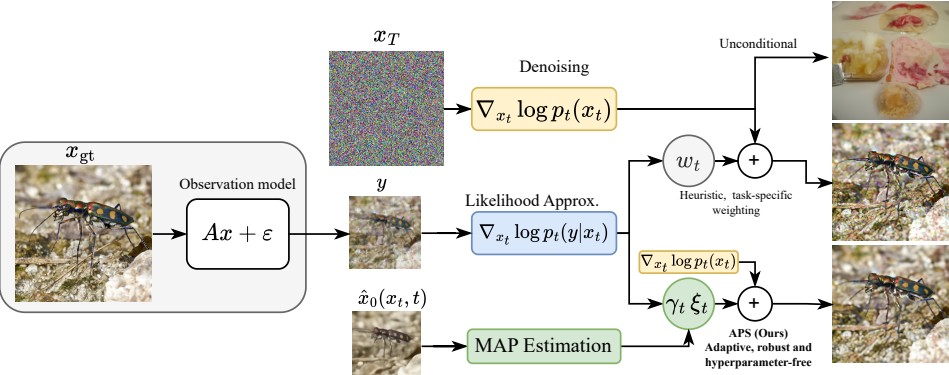

Figure 1: **Schematic overview of our method.** We introduce a principled, hyperparameter-free rule for balancing likelihood guidance with the prior in diffusion-based posterior sampling.

### 3.2.1 APS Update

Let $\epsilon_t^* := \mathbb{E}[\epsilon \mid x_t, y]$ denote the MMSE estimate of the posterior noise at time $t$. For brevity, write $\epsilon_{\theta,t} := \epsilon_\theta(x_t, t)$, $g_t := g(y, x_t)$, and $\tilde{\epsilon}_{\theta,t} := \tilde{\epsilon}_\theta(x_t, t, y)$.

We choose $\xi_t$ so that $\tilde{\epsilon}_\theta = \epsilon_{\theta,t} + \xi_t \, g_t$ (Eq. 13) best approximates $\epsilon_t^*$ in least squares:

$$\xi_t^* := \arg\min_{\xi_t} \left\| \epsilon_t^* - \epsilon_{\theta,t} - \xi_t \, g_t \right\|_2^2 \implies \xi_t^* = \frac{\langle d_t, \, g_t \rangle}{\|g_t\|_2^2}, \qquad d_t := \epsilon_t^* - \epsilon_{\theta,t}, \tag{15}$$

with $\langle \cdot, \cdot \rangle$ the Euclidean inner product (if $\|g_t\|_2 = 0$, set $\xi_t^* = 0$). Substituting $\xi_t^*$ into Eq. 14 yields the APS update:

$$x_{t-1} = \text{DDIM}(x_t) - \gamma_t \frac{\langle d_t, \, g_t \rangle}{\|g_t\|_2^2} g_t. \tag{16}$$

Equivalently, in norm–alignment form,

$$x_{t-1} = \text{DDIM}(x_t) - \gamma_t \|d_t\|_2 \left\langle \hat{d}_t, \, \hat{g}_t \right\rangle \hat{g}_t, \qquad \hat{d}_t := d_t / \|d_t\|_2, \quad \hat{g}_t := g_t / \|g_t\|_2. \tag{17}$$

Thus, APS scales the likelihood step by the residual magnitude $\|d_t\|_2$ and the *alignment* between $\hat{d}_t$ and the chosen likelihood direction $\hat{g}_t$. In practice, we regularize the step size using the measurement-consistent residual noise $\epsilon_t^* - \epsilon_{\theta,t}$: when alignment is strong, the two likelihood surrogates agree and larger steps are warranted; when misaligned, moving along $\hat{g}_t$ risks injecting erroneous guidance, so the step is attenuated. Importantly, the DDIM schedule coefficient $\gamma_t$ is preserved, ensuring guidance scales appropriately under time re-spacing and stochasticity.

However, because $d_t$ and $g_t$ are distinct surrogates of the likelihood update, perfect alignment is unlikely; the projection coefficient $\langle \hat{d}_t, \hat{g}_t \rangle \leq 1$ therefore systematically shrinks the update step, even when the directions largely agree. To counter this, we apply a simple, data-agnostic bias correction, scaling by $2 \approx 1/\mathbb{E}[\langle \hat{d}_t, \hat{g}_t \rangle]$ (empirically $\approx 0.5$ at mid-trajectory), which restores the intended step magnitude. This yields our final update:

$$x_{t-1} = \text{DDIM}(x_t) - \gamma_t \frac{2\langle d_t, g_t \rangle}{\|g_t\|_2^2} g_t = \text{DDIM}(x_t) - 2\gamma_t \|d_t\|_2 \langle \hat{d}_t, \hat{g}_t \rangle \hat{g}_t \qquad (18)$$

Because computing $g_t$ already requires backpropagation through the denoiser, we next show how to evaluate $d_t$ directly in noise space without the denoiser's Jacobian. We discuss this approach in detail in Section A.1.2 of the appendix.

### 3.2.2 EFFICIENT MAP SURROGATE FOR $d_t$

Because $\epsilon_t^*$ is unknown, we approximate it via a maximum a posteriori (MAP) estimate for $x_0$ conditional on $(x_t, y)$. From Bayes' rule,

$$\Phi(x_0) := -\log p(x_0|x_t, y) = -\log p(x_0|x_t) - \log p(y|x_0) + C, \qquad (19)$$

where $C$ encapsulate terms that do not depend on $x_0$. Following prior work (Song et al., 2023; Boys et al., 2023), we adopt

$$p(x_0|x_t) \approx \mathcal{N}(\hat{x}_0, r_t^2 I), \qquad r_t^2 = 1 - \bar{\alpha}_t, \qquad (20)$$

and the linear-Gaussian likelihood Eq. 2. Under these assumptions, the negative log-posterior obeys

$$\Phi(x_0) \propto \frac{1}{2\sigma_y^2} \|y - Ax_0\|_2^2 + \frac{1}{2r_t^2} \|x_0 - \hat{x}_0\|_2^2. \qquad (21)$$

In the linear-Gaussian setting, the posterior mean is the MAP minimizer. Since $\Phi$ is strictly convex, the optimum $x_0^*$ satisfies $\nabla\Phi(x_0^*) = 0$, giving

$$x_0^* = \hat{x}_0 - A^\top \left(AA^\top + \frac{\sigma_y^2}{r_t^2} I\right)^{-1} (A\hat{x}_0 - y). \qquad (22)$$

See full derivation in Section A.1.1 of the appendix. Mapping $x_0^*$ back to the VP noise variable via $x_t = \sqrt{\bar{\alpha}_t}\, x_0 + \sqrt{1 - \bar{\alpha}_t}\, \epsilon$ yields

$$d_t = \epsilon_t^* - \epsilon_{\theta,t} = \frac{\sqrt{\bar{\alpha}_t}}{\sqrt{1 - \bar{\alpha}_t}} A^\top \left(AA^\top + \frac{\sigma_y^2}{r_t^2} I\right)^{-1} (A\hat{x}_0 - y). \qquad (23)$$

**Remark 1** *For many linear operators $A$, the operator $\left(AA^\top + \frac{\sigma_y^2}{r_t^2}I\right)^{-1}$ can be implemented efficiently: exactly via SVD (for small dense $A$), via FFT diagonalization for circulant models (e.g., in super-resolution and deblurring), or via conjugate gradients in the general case.*

The construction of $d_t$ is reminiscent of the residual-based likelihood approximation of (Li & Wang, 2025), but under our Gaussian assumptions and linear $A$, the objective in 21 is minimized exactly.

## 4 EXPERIMENTAL RESULTS

### 4.1 EXPERIMENTAL SETUP

**Tasks and datasets.** We test our method on the CelebA-HQ and ImageNet-256 validation sets, with backbone denoisers trained by Lugmayr et al. (2022) and Dhariwal & Nichol (2021), respectively. Our evaluation is performed by conducting several key image restoration tasks, used also in previous works (Kawar et al., 2022; Zhu et al., 2023): (i) *Super-resolution* $\times 4$ with a bicubic downsampling kernel, in both noiseless and noisy settings; (ii) *Gaussian deblurring* with a $5\times5$ Gaussian kernel (standard deviation 10); and (iii) *Motion deblurring* with randomized $61\times61$ kernels of intensity 0.5, generated using the public implementation.[1] Unless noted otherwise, we add zero-mean i.i.d. Gaussian noise with $\sigma_y = 0.05$, conventionally expressed in $[0,1]$ intensity units. Since we normalize images to the range $[-1, 1]$, we accordingly multiply the noise level by a factor of two.

---

[1] https://github.com/LeviBorodenko/motionblur

**Baselines.** We compare against DDRM (Kawar et al., 2022), DPS (Chung et al., 2023), Diff-PIR (Zhu et al., 2023), DDPG (Garber & Tirer, 2024), ΠGDM (Song et al., 2023), and DSG (Yang et al., 2024). All evaluations use the same datasets, seeds, and implementations of the measure-ment operators. We report PSNR and LPIPS (distortion and perceptual quality), averaged over 1K samples per dataset; for LPIPS, we use the AlexNet variant. To fully assess our step-size strategy (APS), we report results when APS computes the likelihood score using either ΠGDM or DPS (i.e., different choices for $g_t$).

**Sampling details.** We use a DDIM sampler with $\eta = 1$ (i.e., DDPM-equivalent) and 100 diffusion steps. Unless noted otherwise, all baselines also use 100 steps. Exceptions are DPS, which requires 1,000 steps, and DSG, which likewise uses 1,000 steps for ImageNet-256. Although DDRM (Kawar et al., 2022) typically operates with ~20 steps, we run it with 100 for a fair comparison.

## 4.2 COMPARISON WITH OTHER METHODS

Table 1 compares APS with representative posterior samplers built on unconditional diffusion priors (see Section 4.1). Across both datasets, APS delivers near state-of-the-art perceptual quality, attain-ing best or second-best LPIPS in all settings. At the same time, it maintains strong PSNR, incurring only a modest distortion cost, consistent with the perception–distortion trade-off (Blau & Michaeli, 2018). Notably, several PSNR-oriented baselines sharply sacrifice LPIPS, especially on noisy tasks, yielding visibly blurrier reconstructions (e.g., DDPG on SR×4 with $\sigma_y = 0.05$). In contrast, APS remains competitive on *both* metrics across tasks and datasets.

Table 1: Super-resolution / deblurring on **CelebA-HQ** and **ImageNet-256**: PSNR [dB] (↑) / LPIPS (↓). Best results are in **bold**; second-best are underlined. N/A = DDRM inapplicable for non-SVD tasks. Values in gray are excluded as they were obtained with a larger number of sampling steps.

| | CelebA-HQ | | | | ImageNet-256 | | | |
|---|---|---|---|---|---|---|---|---|
| **Method** | Bicub. SR×4 $\sigma_y = 0$ | Bicub. SR×4 $\sigma_y = 0.05$ | Gauss. Deb. $\sigma_y = 0.05$ | Motion Deb. $\sigma_y = 0.05$ | Bicub. SR×4 $\sigma_y = 0$ | Bicub. SR×4 $\sigma_y = 0.05$ | Gauss. Deb. $\sigma_y = 0.05$ | Motion Deb. $\sigma_y = 0.05$ |
| DDRM | **31.64** / 0.054 | 29.26 / 0.090 | **30.53** / 0.074 | N/A | 27.38 / 0.270 | 25.54 / 0.333 | 27.71 / 0.243 | N/A |
| DiffPIR | 30.26 / 0.051 | 27.44 / 0.085 | 28.89 / 0.074 | 27.96 / 0.102 | 26.99 / 0.225 | 24.65 / 0.318 | 26.64 / 0.240 | 25.34 / 0.284 |
| DDPG | 31.60 / 0.052 | **29.39** / 0.105 | 30.41 / 0.068 | **29.02** / 0.082 | **27.41** / 0.255 | **25.55** / 0.354 | **27.73** / 0.205 | **25.94** / 0.249 |
| DSG[†] | 30.40 / 0.051 | 27.57 / 0.072 | 30.29 / **0.051** | 27.57 / 0.079 | 26.08 / 0.198 | 24.33 / 0.203 | 26.69 / 0.153 | 24.33 / 0.197 |
| DPS[†] | 29.39 / 0.065 | 27.49 / 0.086 | 27.75 / 0.084 | 19.63 / 0.227 | 25.56 / 0.236 | 24.05 / 0.271 | 23.59 / 0.294 | 17.52 / 0.468 |
| ΠGDM | 30.93 / **0.038** | 27.23 / 0.078 | 27.67 / 0.087 | 26.15 / 0.104 | 26.72 / **0.122** | 22.83 / **0.227** | 22.85 / 0.268 | 20.97 / 0.292 |
| APS-DPS (Ours) | 30.83 / 0.042 | 28.10 / **0.064** | 30.03 / 0.056 | 27.41 / 0.077 | 27.23 / 0.202 | 25.13 / 0.259 | 26.38 / 0.259 | 23.61 / 0.298 |
| APS-ΠGDM (Ours) | 30.91 / 0.040 | 28.00 / 0.065 | 29.39 / 0.054 | **28.00** / **0.067** | 27.24 / 0.181 | 25.04 / 0.240 | 27.14 / **0.147** | 25.36 / **0.193** |

[†] Methods evaluated with 1K NFEs. On CelebA-HQ, DSG was applied with 100 steps.

## 4.3 ABLATION STUDIES

In this section, we analyze various aspects in our proposed approach. Experiments in this section are carried out by solving SR×4 with $\sigma_y = 0.05$ on 100 samples from the ImageNet-256 validation set, unless specified otherwise.

**Isolating the impact of our adaptive $\xi_t$.** We assess the contribution of the adaptive factor by freezing it to con-stants: $\xi_t = 1$ (replacing $\frac{2\langle d_t, g_t \rangle}{\|g_t\|_2^2}$) and $\xi_t = 2$ to decouple the factor-of-two bias correction from true adaptivity. As shown in Table 2, the adaptive $\xi_t$ consistently outperforms either choice for both APS–ΠGDM and APS–DPS, indicat-ing that alignment-aware step-size modulation is essential.

| $\xi_t$ | APS–PGDM | APS–DPS |
|---|---|---|
| 1 | 24.40 / 0.356 | 23.48 / 0.342 |
| 2 | 22.54 / 0.500 | 20.90 / 0.460 |
| Ours | 24.64 / 0.246 | 24.74 / 0.266 |

Table 2: Impact of $\xi_t$ (PSNR / LPIPS).

**Direct Comparison to DPS.** Across both CelebA-HQ and ImageNet-256, DPS uses ten times more sampling steps than its APS counterpart yet underperforms, even though both use the same likelihood-gradient surrogate $g_t(y, x_t)$ (compare DPS and APS-DPS in Table 1).

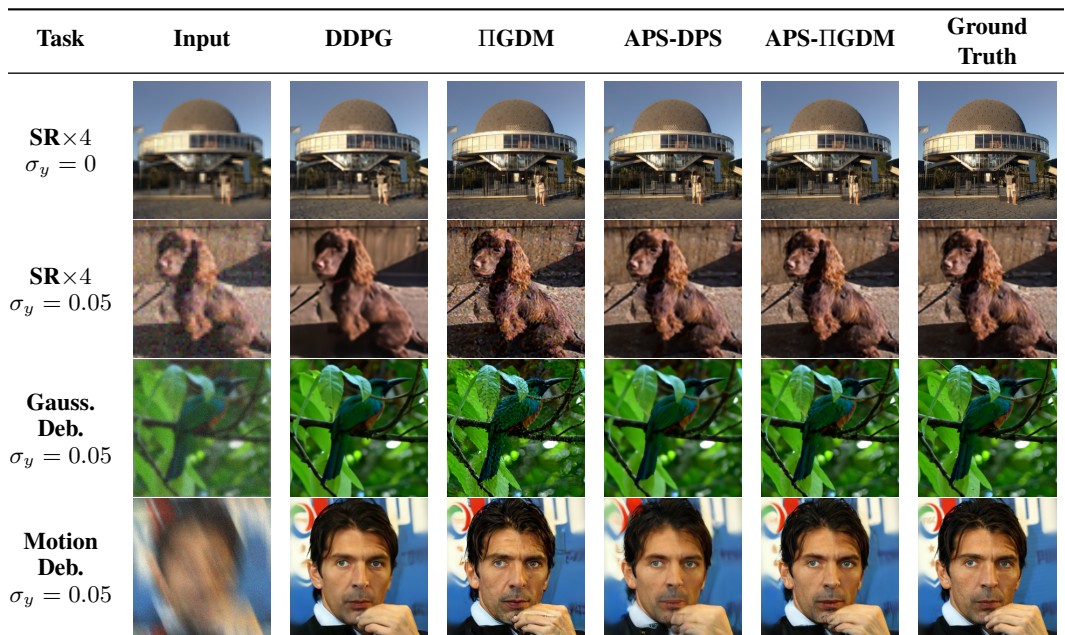

Figure 2: **Qualitative comparison of APS and representative methods.** Best viewed in zoom-in.

**Comparison to ΠGDM.** Beyond the likelihood approximation, Song et al. (2023) introduce a time-decaying multiplicative step size equal to $(1-\bar{\alpha}_t)$. While this choice performs well at 100 sampling steps, it does not account for the diffusion schedule's discretization (i.e., changing the number of steps). Consequently—and counter-intuitively—*increasing* the number of steps degrades performance in both PSNR and LPIPS, rather than improving it, as was also observed by Mardani et al. (2023). In contrast, our method explicitly incorporates step spacing through $\gamma_t$, yielding a scalable sampler whose perceptual quality improves with more steps, with only negligible PSNR deterioration (Figs. 3, 4).

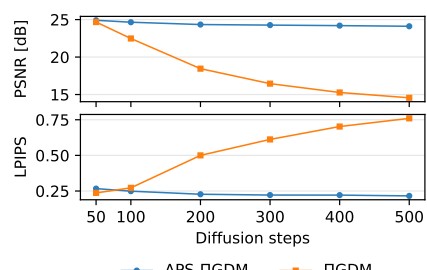

Figure 3: Comparison of APS vs. ΠGDM as a function of sampling steps.

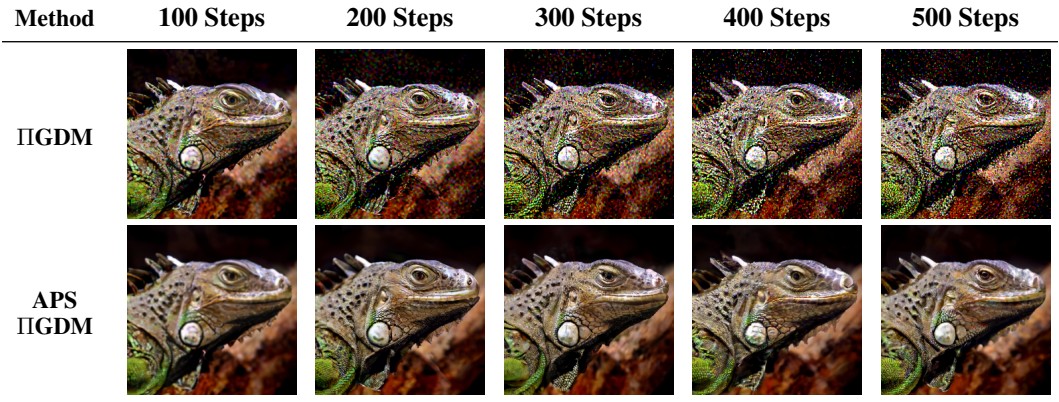

Figure 4: **Qualitative comparison of APS vs. ΠGDM.** SR×4 with $\sigma_y = 0.05$. Reconstructions of the same image using varying numbers of sampling steps. As the step count increases, Π-GDM deteriorates due to measurement-noise leakage into the reconstructions, whereas APS improves.

**Measurement noise level.** In many posterior sampling schemes, increasing the observation noise not only removes information but can also *leak* through the measurement-consistency gradients into the reconstruction. We evaluate SR×4 across increasing $\sigma_y$ and compare APS to ΠGDM using PSNR and LPIPS. Figure 5 shows that APS mitigates the inevitable degradation, with quality declining roughly linearly as the noise grows, whereas ΠGDM exhibits a markedly sharper drop. We exclude other baselines from this ablation because they require retuning hyperparameters for each noise level (e.g., DDPG (Garber & Tirer, 2024)).

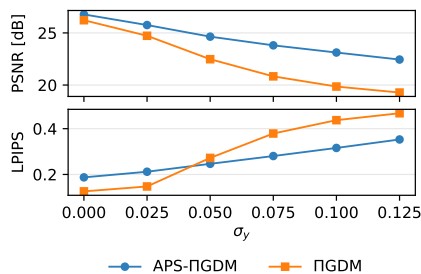

Figure 5: Response to increasing measurement noise.

**Design choices for $g_t$ and $d_t$.** Beyond our default role split (direction from $g_t$, magnitude from $d_t$), we also evaluate alternative pairings, including settings with $d_t = g_t$ and a symmetric variant that averages the two, $(g_t + d_t)/2$, as detailed in Appendix A.2.2. Our ablations show that using the MAP-based surrogate for $d_t$ is not only negligible in runtime but also outperforms competing choices. We find that omitting the Jacobian in the MAP-based magnitude yields more stable step sizes, whereas incorporating Jacobian information in the *direction* is crucial. Further discussion appears in Appendix A.1.2.

## 5 CONCLUSION

This paper introduces **APS**, a robust, task-agnostic and hyperparameter-free strategy for setting the guidance scale of the likelihood term when combined with a pretrained unconditional diffusion prior. Our method surpasses leading approaches in perceptual quality and provides high PSNR, while remaining scalable and adaptive across settings and diffusion schedules.

**Limitations.** Our current derivation entails several limitations that motivate further research. First, the analysis assumes a linear forward model, which guarantees that the posterior mean minimizes Eq. 19 and enables fast evaluation of $d_t$ via Eq. 23. Extending APS to *nonlinear* operators will require principled surrogates or approximations of the posterior update under nonlinearity. Second, diffusion in high-resolution pixel space is resource-intensive, which has motivated the use of latent diffusion models (LDMs) that operate in a compressed latent space. However, posterior sampling becomes challenging when measurements live in the image domain, as the latent decoder must be involved in the measurement-consistency term (see, e.g., (Rout et al., 2023; Song et al., 2024)). Adapting APS to latent spaces is therefore an important direction for future work.

**Outlook.** Despite these limitations, APS achieves state-of-the-art reconstructions on prominent imaging tasks by providing a principled recipe for setting the likelihood step size in posterior samplers, building on existing likelihood approximations. Our derivations reduce reliance on ad hoc hyperparameter tuning while preserving proper scaling with respect to the number of steps, respacing, and stochasticity in the diffusion process. We believe APS offers a solid foundation for broader posterior-guided sampling methods in both pixel and latent domains.

Our code will be released upon acceptance.

USE OF LARGE LANGUAGE MODELS FOR WRITING

We acknowledge the use of large language models to assist with typographical corrections, phrasing, and self-review aimed at improving the clarity and structure of this manuscript.

REPRODUCIBILITY STATEMENT

All of our work is reproducible: we detail all parameters and use publicly available datasets. An additional section on reproducibility appears in the appendix. Furthermore, we will release our code upon acceptance.

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

# A APPENDIX

## A.1 ADDITIONAL METHOD DETAILS

### A.1.1 PROOF OF EQ. 22

Let $\Phi(x_0)$ be as in Eq. 19. Since $\Phi$ is convex, the optimum $x_0^*$ satisfies

$$\nabla_{x_0}\Phi(x_0^*) \;=\; 0 \;\iff\; \sigma_y^{-2}A^\top(Ax_0^* - y) \;+\; r_t^{-2}(x_0^* - \hat{x}_0) \;=\; 0.$$

Proceeding step by step,

$$\sigma_y^{-2}\,A^\top(Ax_0^* - y) \;+\; r_t^{-2}\,(x_0^* - \hat{x}_0) \;=\; 0$$

$$\iff x_0^* - \hat{x}_0 \;=\; \frac{r_t^2}{\sigma_y^2}\,A^\top(y - Ax_0^*)$$

$$\iff \left(\tfrac{r_t^2}{\sigma_y^2}\,A^\top A + I_n\right)x_0^* \;=\; \hat{x}_0 \;+\; \tfrac{r_t^2}{\sigma_y^2}\,A^\top y$$

$$\iff \left(A^\top A + \tfrac{\sigma_y^2}{r_t^2}\,I_n\right)x_0^* \;=\; \tfrac{\sigma_y^2}{r_t^2}\,\hat{x}_0 \;+\; A^\top y$$

$$\iff \left(A^\top A + \tfrac{\sigma_y^2}{r_t^2}\,I_n\right)x_0^* \;=\; A^\top A\,\hat{x}_0 \;+\; \tfrac{\sigma_y^2}{r_t^2}\,\hat{x}_0 \;+\; A^\top y \;-\; A^\top A\,\hat{x}_0$$

$$\iff \left(A^\top A + \tfrac{\sigma_y^2}{r_t^2}\,I_n\right)x_0^* \;=\; \left(A^\top A + \tfrac{\sigma_y^2}{r_t^2}\,I_n\right)\hat{x}_0 \;+\; A^\top(y - A\hat{x}_0)$$

$$\iff x_0^* \;=\; \hat{x}_0 \;+\; \left(A^\top A + \tfrac{\sigma_y^2}{r_t^2}\,I_n\right)^{-1}A^\top(y - A\hat{x}_0). \tag{24}$$

Using the "push-through" identity

$$\left(A^\top A + \lambda I_n\right)^{-1}A^\top \;=\; A^\top\left(AA^\top + \lambda I_m\right)^{-1} \qquad \forall \lambda > 0,$$

Eq. 24 is equivalently

$$x_0^* \;=\; \hat{x}_0 \;+\; A^\top\left(AA^\top + \tfrac{\sigma_y^2}{r_t^2}I_m\right)^{-1}(y - A\hat{x}_0), \tag{25}$$

which can be preferable in memory when $m < n$ (e.g. super-resolution).

### A.1.2 ON JACOBIAN–FREE APPROXIMATIONS

In Section 3.2.2 we proposed a MAP-based likelihood surrogate computed with respect to the prior mean $\hat{x}_0$, thereby obviating the need to evaluate the Jacobian $J_t := \partial\hat{x}_0/\partial x_t$, which would otherwise require backpropagating through the denoiser at every step. This design trades exactness for efficiency: strictly speaking, the correct conditional score with respect to $x_t$ *does* involve $J_t$ via the chain rule. Indeed, for any differentiable functional $L(\hat{x}_0)$,

$$\nabla_{x_t}L(\hat{x}_0) \;=\; J_t^\top\,\nabla_{\hat{x}_0}L(\hat{x}_0). \tag{26}$$

The impact of neglecting $J_t$ has been examined in prior work. For example, Chung et al. (2024) identify conditions under which $J_t$ can be replaced by a low-cost operation, while Poole et al. (2022) observed that omitting $J_t$ can simplify and *stabilize* optimization, particularly in low-noise regimes (both findings are consistent with our experience). From a geometric standpoint, $J_t$ may induce an anisotropic linear transform that can *rotate* and *rescale* the vector $\nabla_{\hat{x}_0}L(\hat{x}_0)$; including it everywhere may unintentionally overweight directions amplified by $J_t$, whereas discarding it everywhere can underrepresent how changes in $x_t$ influence the measurement-consistency objective.

Our scheme strikes a practical balance. APS cleanly separates *magnitude* and *direction*: the step *size* is governed by $d_t$ (a MAP-based residual that is computed in $\hat{x}_0$-space without $J_t$), while the *direction* is provided by $g_t$, which may incorporate $J_t$ when a Jacobian-aware likelihood surrogate is used (e.g., DPS or ΠGDM). In other words, we avoid injecting the anisotropy of $J_t$ into the *scale* of

the update—mitigating over/under-shoot due to ill-conditioning—yet we still allow $J_t$ to influence the *direction* through $g_t$ when this is available and beneficial.

A further benefit is architectural agnosticism: the MAP-based $d_t$ depends only on the measurement model and on $\hat{x}_0$, not on the particular score-parameterization. Consequently, the same construction applies across Variance–Preserving (VP), Variance–Exploding (VE), or probability-flow/flow-based samplers, and is thus future-compatible with alternative priors.

In summary, while omitting $J_t$ is theoretically inexact, using a Jacobian-free magnitude ($d_t$) together with a potentially Jacobian-aware direction ($g_t$) yields an effective and stable compromise that preserves computational tractability at high resolution and integrates cleanly with existing likelihood surrogates ((Chung et al., 2024; Poole et al., 2022)).

For completeness, we include an ablation in Section A.2.2 that examines the impact of alternative surrogate choices for both $d_t$ and $g_t$.

## A.2  ADDITIONAL RESULTS

### A.2.1  ADDITIONAL ABLATIONS

Stochasticity. Many posterior samplers set $\eta = 1$ (see Section 2.2), allowing fresh noise to mitigate artifacts that arise when enforcing consistency with noisy observations. We evaluate APS across a range of $\eta$ values and compare against ΠGDM. Figure 6 shows that, whereas ΠGDM benefits primarily from highly stochastic updates (large $\eta$), APS exhibits markedly weaker dependence on $\eta$, maintaining similar performance across stochasticity levels.

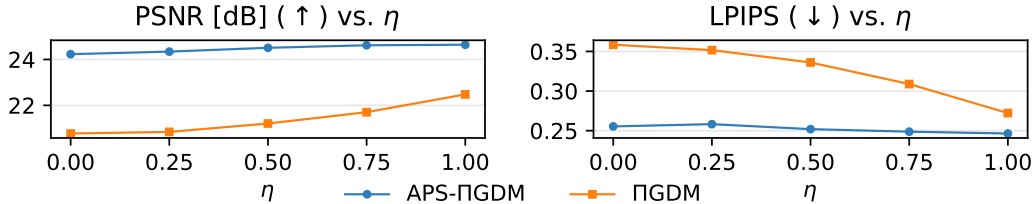

Figure 6: **Sensitivity to stochasticity ($\eta$).** PSNR/LPIPS for SR$\times$4 as a function of $\eta$. APS maintains similar performance across stochasticity levels, whereas ΠGDM is more sensitive to the amount of injected noise.

### A.2.2  DIFFERENT CHOICES FOR $g_t$ AND $d_t$

Our method combines two complementary surrogates to regularize posterior updates: the *direction* is set by $g_t$, and the *magnitude* by $d_t$, with a correlation-based correction that attenuates risky steps when the two disagree. It is nevertheless natural to consider alternative assignments. In this section we vary both ingredients by choosing

$$(g_t, d_t) \in \{\texttt{dps}, \texttt{pgdm}, \texttt{map}\} \times \{\texttt{dps}, \texttt{pgdm}, \texttt{map}\},$$

and, in addition, evaluate a symmetric variant that steps along the average direction $(g_t + d_t)/2$ (thereby sharing both role and responsibility between the two surrogates). We evaluate each combination on 100 images from the ImageNet-256 validation set for SR$\times$4 with $\sigma_y = 0.05$.

The results of our experiments are given in Table 3.

For clarity, the three surrogates used to instantiate $\tilde{\epsilon}_\theta(x_t, t, y)$ are:

$$\tilde{\epsilon}_\theta^{\Pi\text{GDM}}(x_t, t, y) = \epsilon_\theta(x_t, t) + \frac{\bar{\alpha}_t}{r_t^2}\left(\frac{\partial \hat{x}_0}{\partial x_t}\right)^\top A^\top\left(AA^\top + \frac{\sigma_y^2}{r_t^2}I\right)^{-1}(y - A\hat{x}_0), \tag{27}$$

$$\tilde{\epsilon}_\theta^{\text{DPS}}(x_t, t, y) = \epsilon_\theta(x_t, t) + \bar{\alpha}_t\left(\frac{\partial \hat{x}_0}{\partial x_t}\right)^\top A^\top(y - A\hat{x}_0), \tag{28}$$

$$\tilde{\epsilon}_\theta^{\text{MAP}}(x_t, t, y) = \epsilon_\theta(x_t, t) + \frac{\sqrt{\bar{\alpha}_t}}{\sqrt{1 - \bar{\alpha}_t}}A^\top\left(AA^\top + \frac{\sigma_y^2}{r_t^2}I\right)^{-1}(A\hat{x}_0 - y). \tag{29}$$

| $g_t$ | $d_t$ | PSNR [dB] ($\uparrow$) | LPIPS ($\downarrow$) | Time/Image [s][†] |
|---|---|---|---|---|
| DPS | DPS | 23.43 | 0.340 | 9.4 |
| DPS | ΠGDM | 22.63 | 0.501 | 17.8 |
| DPS | MAP | **24.72** | 0.266 | 9.5 |
| ΠGDM | DPS | 20.42 | 0.445 | 18.0 |
| ΠGDM | ΠGDM | 24.43 | 0.354 | 9.4 |
| ΠGDM | MAP | 24.65 | **0.249** | 9.5 |
| MAP | DPS | 20.42 | 0.445 | 9.8 |
| MAP | ΠGDM | 21.04 | 0.576 | 9.5 |
| MAP | MAP | 22.14 | 0.399 | 5.3 |

| $(g_t + d_t)/2$ | PSNR [dB] ($\uparrow$) | LPIPS ($\downarrow$) | Time/Image [s][†] |
|---|---|---|---|
| ΠGDM , DPS | 24.02 | 0.335 | 18.0 |
| MAP , ΠGDM | 22.84 | 0.426 | 9.5 |
| DPS , MAP | 22.16 | 0.435 | 9.7 |

Table 3: Ablation over pairings of direction ($g_t$) and magnitude ($d_t$), and the averaged.

[†] Experiments were run on NVIDIA L40S GPUs. Runtimes depend on hardware and settings, so values should be interpreted comparatively rather than absolutely.

## A.3 REPRODUCIBILITY

**ΠGDM implementation details.** As no official implementation of ΠGDM is publicly available, we reimplemented it based on the description in (Song et al., 2023) and the publicly released RED-diff code (Mardani et al., 2023)[2]. Note that our notation (and code) follow the VP–DDPM parameterization, whereas both Song et al. (2023) and Mardani et al. (2023) use a Variance–Exploding (VE) formulation. To verify correctness, we cross-validated our reimplementation against results reported in Song et al. (2023) on overlapping setups. In particular, both implementations yield an LPIPS of $0.122$ on clean bicubic SR$\times 4$, confirming consistency. Notice that when using any likelihood surrogate for $g_t$, it is normalized, removing any time-constants scaling.

Our code will be released upon acceptance.

---

[2]https://github.com/NVlabs/RED-diff/blob/master/algos/pgdm.py

## A.4 ADDITIONAL VISUAL RESULTS

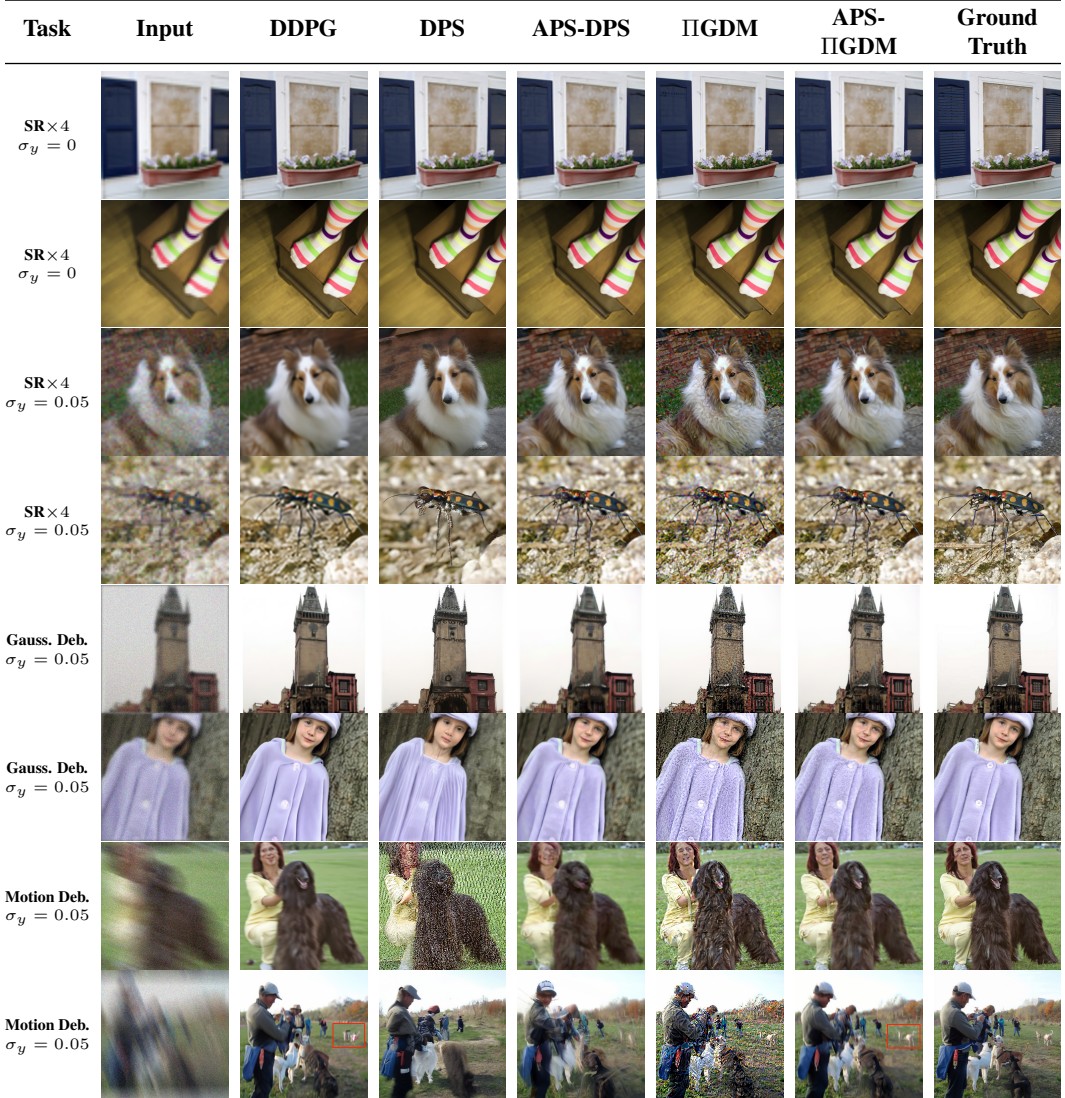

Figure 7: Additional visual results on **ImageNet-256** validation set. Best viewed in zoom-in.

