# OpenReview forum: "Adaptive Guidance Scaling for Posterior Diffusion-based Sampling"
_ICLR.cc/2026/Conference — ICLR 2026 Conference Withdrawn Submission_

### Official Review · Reviewer_Cc9K · 2025-10-28

**Soundness:** 3
**Presentation:** 3
**Contribution:** 2
**Rating:** 4
**Confidence:** 3

**Summary:**

The authors present a hyperparameter-free approach to find the step size for conditional diffusion sampling methods such as DPS [1].
They apply their approach "Adaptive Posterior Diffusion Sampling" (APS) to several linear inverse problems on both CelebA and ImageNet.

[1] Chung et al. "Diffusion Posterior Sampling for General Noisy Inverse Problems" (2023)

**Strengths:**

- It is fairly well known that DPS can be unstable, so it is a good idea to develop a better choice for the step size/guidance strength.
- For specific linear inverse problems the MAP surrogate can be efficiently evaluated
- In all the experiments in Table 1 APS-DPS performs better than DPS and in most cases APS-$\Pi$GDM performs better than $\Pi$GDM

**Weaknesses:**

- The current presentation is limited to linear inverse problems with additive Gaussian noise. For non-linear forward operators or other noise types the surrogate for $d_t$ (Section 3.2.2) will become computationally expensive.
- The calculation of $d_t$ requires to solve a linear system involving the forward operator $A$ at every sampling step. This might be feasible for certain types of forward operators (e.g., super-resolution or deblurring), but this
becomes expensive is the forward operator itself is more costly to evaluate (e.g., computed tomography)

**Questions:**

- The difference in the results in Table 2 between $\xi_t=1$ and your method seem to be minor. Does this mean incorporating the $\gamma_t$ in (18) has more importance?
- Why is the bias correction needed? So, where do you introduce the bias?
- Can you provide a plot of change of the step size $\| d_t \|_2$ over sampling iterations?
- Can you add a comparison of the sampling time (as your method needs to compute $d_t$ at every iteration)?
- How would this approach translate to latent diffusion models?

---

### Official Review · Reviewer_n43G · 2025-10-31

**Soundness:** 3
**Presentation:** 2
**Contribution:** 3
**Rating:** 4
**Confidence:** 5

**Summary:**

The paper proposes an adaptive likelihood stepsize estimation strategy for DPS and PGDM. The algorithm is tested on superresolution and deblurring tasks. The overall idea is interesting, but execution may be improved.

**Strengths:**

- I really like the overall idea of adaptively changing the step size based on existing iterates and residuals.
- The adaptive algorithm shows good performance.

**Weaknesses:**

- Like I said, I like the overall idea. But the main step that went into it is not clearly justified. Why is one trying to align the estimates toward the MMSE estimate of the posterior given x_t, y? This is going to be a bad estimate especially for high values of t (i.e. at higher noise level). There is no clear justification given for the objective in (15).
- Similarly, there is a hand-wavy argument about doubling the scaling in (18), but this is neither justified nor empirically demonstrated.
- The MAP surrogate is also not discussed in detail. Why is this a good solution? How does the solution quality change along the trajectory?
- In terms of experiments, all problems considered are resolution related. Please include results from non-resolution related tasks for random (~90%) and box (128 \times 128) inpainting.
- It is unclear how many test datasets were used for the results reported. Usually a 1000 images are reported in these experiments.

Minor:
- In (1), please use matrix notation, as this idea only works for linear forward operators, and not non-linear ones.
- It is not fair to say that other methods directly add \xi_t g(y,x_t), since they define their \xi_t to be equivalent to this paper's \gamma_t \xi_t.
- LPIPS is typically reported using VGG (e.g. in DPS).
- I'm not sure why the authors are saying PGDM is not publicly available. This is available in the RED-Diff repository. Also the VP-SDE to VE-SDE conversion is already covered in the PGDM paper Appendix A.1, this is a matter of scaling by \sqrt{\bar{\alpha}_t}.
- The authors highlight that they incorporate likelihood gradients into a DDIM sampler in a principled way as one of their three main contributions. However, the switch from Eq. 10 to 13 brings an additional factor of \xi_t, which is not present in Eq. 10. I know this is used in all other works, but the inclusion of this variable hinders the derivation.

**Questions:**

- What is the justification for trying to align the estimates toward the MMSE estimate of the posterior given x_t, y, i.e. the objective in (15)?
- What is the justification for the scaling in (18)?
- How good is the MAP solution for different parts of the trajectory?
- Can you please include other relevant inverse problems that are not focusing on resolution improement, such as random (~90%) and box (128 \times 128) inpainting?
- How many test datasets were used in the reported results?
- Why was LPIPS calculated using AlexNet instead of VGG, as in DPS or PGDM?

I'm willing to increase my score if the authors address these issues sufficiently

---

### Official Review · Reviewer_eyDA · 2025-10-31

**Soundness:** 2
**Presentation:** 3
**Contribution:** 2
**Rating:** 2
**Confidence:** 4

**Summary:**

The paper proposes Adaptive Posterior diffusion Sampling (APS), a method designed to automatically adjust the guidance strength in diffusion-based inverse problem solvers. Instead of using a manually tuned hyperparameter, APS introduces an adaptive scaling factor that depends on the alignment between the denoiser’s prediction and a likelihood-based correction term. The method is positioned as a hyperparameter-free alternative to approaches like DPS and ΠGDM.

**Strengths:**

1. The motivation to remove manual tuning of the guidance strength is clear and relevant to practical applications.
2. Writing quality and presentation are good, making the method easy to follow.
3. The paper provides meaningful ablations and visual examples to support its claims of reduced sensitivity to hyperparameters.

**Weaknesses:**

1. The paper makes several simplifying assumptions that reduce mathematical rigor, the most notable being the elimination of the Jacobian term as discussed in Appendix A.1.2. The authors justify this choice by citing the DDS paper, which also omits Jacobians. However, DDS operates under a different geometric framework where the denoising dynamics are modeled along a locally affine approximation of the data manifold and the Tweedie denoised estimate is updated before each DDIM sampling step. In contrast, APS removes the Jacobian without adopting a comparable manifold assumption or analyzing how this omission affects posterior consistency. It is therefore unclear whether the Jacobian-free simplification remains theoretically valid in the DPS or ΠGDM setting.
2. No inpainting task is reported, even though inpainting is a standard benchmark in the inverse problems community. Given the reliance on the no-Jacobian assumption, it is plausible that the method may not perform well on tasks such as inpainting, which are typically more challenging without the Jacobian term. Omitting these results raises concerns about the generality and robustness of the proposed approach.
3. The proposed method is only demonstrated on inverse problems with linear and differentiable measurement operators. It remains unclear how APS would handle non-linear or non-differentiable operators such as JPEG compression, quantization, non-linear deblurring, or phase retrieval.
4. Although APS defines its adaptive scale as $ξ_t^* = \frac{<dt,gt>}{||gt||^2}$, the surrogate posterior correction $d_t$ is itself computed using the MAP estimator. Because both $d_t$ and $g_t$ depend on the same residual term ($\mathbf{A}\hat{\mathbf{x}}_0−\mathbf{y}$), the alignment $<dt,gt>$ effectively measures self-correlation rather than a genuine interaction between prior and likelihood information. Ultimately, it seems to me that the adaptive scale $ξ_t^*$ is not computed from two independent signals.
5. The authors' claim of achieving “state-of-the-art” results is difficult to justify, as Table 1 shows that the proposed APS-DPS and APS-ΠGDM variants fail to consistently outperform even the relatively outdated baselines, achieving the best results in only about 10–15% of cases. Without comparisons to more recent methods (*e.g.*, RED-Diff, MGPS) or stronger baselines (*e.g.*, DDNM), it remains uncertain whether APS offers any real improvement over the current state of the art.
6. The literature review does not acknowledge the ECCV 2024 paper (https://doi.org/10.1007/978-3-031-73010-8_26), which pursues the same overarching goal of adaptive posterior sampling but through a different mechanism. Although the strategy differs from APS, both aim to automate the balance between prior and likelihood terms in diffusion-based inverse problem solvers. Failing to mention this work gives the impression that APS is the first to address adaptive posterior sampling, when in fact related efforts already exist with complementary approaches.

**Minor Comments:**
- Vectors and matrices should be written in boldface to clearly distinguish them from scalar quantities.
- Providing a structured algorithm in the main text would make the proposed method easier to follow and reproduce.
- It is important to report the selected heuristic task-specific weighting ($\omega_t$) used for DPS and ΠGDM in each experimental setup.
- Although the authors state otherwise, ΠGDM code is publicly available (https://github.com/NVlabs/RED-diff/blob/master/algos/pgdm.py).

**Questions:**

1. *Regarding weakness 1:* Have the authors evaluated whether including the Jacobian, as in DPS, leads to any difference in adaptive guidance behavior or reconstruction quality?
2. *Regarding weakness 3:* Since the derivation assumes linearity in the forward operator, how would the MAP surrogate behave when applied to non-linear or non-differentiable operators? Do the authors anticipate that APS would remain stable and effective in these cases?
3. *Regarding weakness 4:* Can the authors clarify whether any component of $d_t$ is statistically independent of the likelihood gradient $g_t$?

---

### Official Review · Reviewer_g99D · 2025-11-10

**Soundness:** 2
**Presentation:** 3
**Contribution:** 2
**Rating:** 4
**Confidence:** 4

**Summary:**

This paper introduces Adaptive Posterior Sampling (APS), a method for dynamically adjusting the guidance scale when incorporating likelihood gradients into diffusion-based inverse problem solvers. The main idea is to adaptively determine the step size of the likelihood update based on the degree of alignment between two surrogate estimates of the posterior gradient: one derived from the denoiser (as in standard score-based models) and another approximated via a MAP formulation.

The approach is designed to be hyperparameter-free, theoretically consistent with DDIM’s time-scaling, and broadly applicable across different inverse problems. APS is evaluated on standard linear inverse tasks such as super-resolution, Gaussian debluring, and motion debluring, using CelebA-HQ and ImageNet-256 datasets. Experiments show improvements in perceptual quality (LPIPS) with competitive or slightly higher PSNR compared to prior methods such as DPS, ΠGDM, and DDRM.

**Strengths:**

1. The paper is well written and technically sound. The derivations connecting Tweedie’s formula, MAP inference, and diffusion posterior updates are logically structured and transparent. The idea of adapting the likelihood step based on local posterior geometry is intuitive and elegantly motivated. The use of a Jacobian-free variant for computational efficiency is practical and well justified.

2. The experiments are actively covering multiple datasets, degradation operators, and noise levels. Comparisons are fair, and ablations on the adaptive factor and surrogate estimators provide valuable insight. The reported results suggest consistent improvements in image perceptual quality with similar or lower computational cost.

**Weaknesses:**

The paper is well written and technically correct, but the main idea feels more like a careful refinement of known methods than a new concept. The adaptive scaling rule is based on a simple idea, which is measuring how well two gradient estimates agree and adjusting the step size accordingly. This is elegant, but not deeply new.

1. The method’s key equation is quite straightforward. It computes how much the denoiser gradient and likelihood gradient align, and then rescales the update based on this alignment. The paper claims this makes sampling more stable, but there is no strong analysis or evidence showing why this is the case. The improvement is more observed than explained.

2. Although the experiments are thorough, the performance gains are relatively small. The LPIPS scores improve slightly and PSNR sometimes decreases. The visual results look good, but not dramatically better than existing methods. Without more insight into why the method works, it feels like a well-tuned variant rather than a major advance.

3. The theoretical development assumes the problem is linear and the noise is Gaussian. This makes the math clean, but it limits the method’s generality. Many real inverse problems are nonlinear or learned (for example, when using a neural forward model), and it is not clear how APS could handle those cases.

**Questions:**

1. Can you give more intuition or evidence for why aligning the denoiser and likelihood gradients helps? For example, does it make the sampling path smoother or reduce noise in the posterior updates?

2. The method assumes a linear and Gaussian forward model. How could APS be extended to nonlinear or learned operators? Would the adaptive scaling still work?

3. Could the adaptive scaling be understood as a form of adaptive learning rate control, similar to how optimizers like Adam adjust step sizes? Would that analogy help explain why it improves stability?

---

### Note · Authors · 2025-11-14

**Comment:**

We sincerely thank the reviewers for the time and effort they dedicated to evaluating our submission. After careful consideration, we have decided to withdraw the paper.

We appreciate the reviewers’ insights and constructive feedback. At the same time, we feel that the central contribution of our work—introducing a simple, clearly-motivated, and hyperparameter-free approach for image reconstruction that requires neither finetuning nor task-specific modifications—may not have been fully recognized in the current submission and the reviews it got. Therefore, we decided to withdraw the paper and resubmit a strengthened version of the paper that better emphasize its unique and important contribution and also take into account the reviewers’ comments.

**Withdrawal Confirmation:**

I have read and agree with the venue's withdrawal policy on behalf of myself and my co-authors.